# Asymmetries of Left and Right Adrenal Glands in Neural Innervation and Glucocorticoids Production

**DOI:** 10.3390/ijms242417456

**Published:** 2023-12-14

**Authors:** Rengui Saxu, Yong Yang, Harvest F. Gu

**Affiliations:** 1Laboratory of Molecular Medicine, School of Basic Medicine and Clinical Pharmacy, China Pharmaceutical University, Nanjing 210009, China; 17320056317@163.com; 2Center for New Drug Safety Evaluation and Research, State Key Laboratory of Natural Medicines, China Pharmaceutical University, Nanjing 211198, China

**Keywords:** adrenal glands, left–right asymmetry, glucocorticoids, Shh, miRNA-466i-5p

## Abstract

The adrenal gland is paired peripheral end organs of the neuroendocrine system and is responsible for producing crucial stress hormones from its two functional compartments, the adrenal cortex, and the adrenal medulla under stimuli. Left–right asymmetry in vertebrates exists from the central nervous system to peripheral paired endocrine glands. The sided difference in the cerebral cortex is extensively investigated, while the knowledge of asymmetry of paired endocrine glands is still poor. The present study aims to investigate the asymmetries of bilateral adrenal glands, which play important roles in stress adaptation and energy homeostasis via steroid hormones produced from the distinct functional zones. Left and right adrenal glands from male C57BL/6J mice were initially histologically analyzed, and high-throughput RNA sequencing was then used to detect the gene transcriptional difference between left and right adrenal glands. Subsequently, the enrichment of functional pathways and ceRNA regulatory work was validated. The results demonstrated that the left adrenal gland had higher tissue mass and levels of energy expenditure, whereas the right adrenal gland appeared to be more potent in glucocorticoid secretion. Further analysis of adrenal stem/progenitor cell markers predicted that Shh signaling might play an important role in the left–right asymmetry of adrenal glands. Of the hub miRNAs, miRNA-466i-5p was identified in the left–right differential innervation of the adrenal glands. Therefore, the present study provides evidence that there are asymmetries between the left and right adrenal glands in glucocorticoid production and neural innervation, in which Shh signaling and miRNA-466i-5p play an important role.

## 1. Introduction

In vertebrates, left–right asymmetry exists from the central structure to the peripheral paired endocrine gland [1]. The sided difference in the central nervous system was extensively investigated, and the asymmetry of paired endocrine glands and the biological significance of this laterality difference are poorly understood. The adrenal gland is paired with the peripheral end organs of the neuroendocrine system and is responsible for producing crucial stress hormones from its two functional compartments, the adrenal cortex and the adrenal medulla under stimuli [2]. The adrenal cortex secretes steroid hormones, glucocorticoids (cortisol in humans and corticosterone in rodents), mineralocorticoids (mainly aldosterone), and androgens, while the adrenal medulla secretes catecholamine, adrenalin, and noradrenalin [3]. The proper functioning of these hormones requires the coordination of both hormonal and neural regulatory processes. Hormonal regulators with adrenal functions mainly include well-known circulating hormones, adrenocorticotropic hormone and angiotensin II, via the independent endocrine feedback loops, the hypothalamus–pituitary–adrenocortical (HPA) axis and the renin–angiotensin–aldosterone system (RAAS), while the neural regulation of adrenal function is performed by neurotransmitters and neuropeptides released from chromaffin cells and different types of neurons innervating the glands [4,5].

In addition to the sided differences in organ weight and venous drainage in the gonads and adrenals, the most evident asymmetry in paired endocrine glands is their neural innervation [6]. In the ovary, bilateral vagotomy resulted in reduced compensatory ovarian hypertrophy only in right unilaterally ovariectomized animals, suggesting a side-specific innervation [1]. In the testis, unilateral orchidectomy caused a sided difference in the RNA synthesizing activity of hypothalamic neurons and the GnRH content of the mediobasal hypothalamus, which suggested a direct neural pathway between the testis and the central nervous system (CNS) [7]. For functional asymmetries in paired endocrine glands, the right testis exhibits a greater response to LH treatment or unilateral orchidectomy than the left testis, and the right half of the hypothalamus contained significantly more GnRH than the left half [8,9]. 

In the adrenal gland, no functional asymmetries were reported in the left and right adrenal, while asymmetric neural innervation in the bilateral adrenal gland has been reported by Gerendai et al. [10]. Holzwarth and Dallman demonstrated that neural-mediated compensatory adrenal growth can be blocked by a lesion on the hypothalamus hemi-islands ipsilateral to the adrenal removed [11]. Then, Toth et al. [12] investigated the predominance of supraspinal innervation in the left adrenal gland compared to the right adrenal one using a viral trans-neuronal tracing technique and suggested the presence of side-specific neurons that project into both organs. Despite the presence of asymmetrical neural innervation being observed, the biological importance of this sided difference and its clinical implications are largely unknown. It is reported that psychiatric disorders, like major depression, which hallmarks the dysfunction of the HPA axis, were found to be associated with variations in asymmetric adrenal regulation [13,14]. In adrenal tumors, unilateral adrenal adenoma causing primary aldosteronism, the most common cause of secondary hypertension, is more commonly detected on the left adrenal gland than on the right side [15]. Understanding the functional and transcriptional differences in left and right adrenal glands can provide evidence to reveal the genetic basis of these diseases. In the present study, C57BL/6J mice were used as an animal model to investigate the morphological, histological, and transcriptional differences in the left and right adrenals. Understanding the laterality difference in the adrenal glands could offer valuable insights into the neuroendocrine asymmetries in the CNS-target endocrine gland system, as well as other paired organs, such as the kidneys.

## 2. Results

### 2.1. Morphology and Histology of Left and Right Adrenal Glands in Mice

To compare the morphological difference in the left and right adrenals, bilateral mouse adrenals were removed and weighted, and morphological images were captured by a body-field microscope Motic SMZ-168 (Motic Germany GmbH, Wetzlar, Germany). As a result, the left adrenal gland tended to be triangle or coma-shaped, while the right adrenal gland was elliptical or wedge-shaped (Appendix A). In a comparison of the organ weight, the left adrenal gland was found to be significantly heavier than the right one (*p* < 0.05) (Figure 1a). To calculate morphological parameters, including the volume of the entire adrenal, adrenal zones, and the ratio, a histological analysis with H&E staining was performed (Appendix A). As shown in Figure 1b,d–f, the volume of the entire adrenal and two major components (cortex, medulla) in the left adrenal gland was larger compared to the right side, and a statistical significance was observed in the entire adrenal volume (*p* < 0.05). The morphological parameters of the area and ratio of different adrenal components were summarized in Table 1a,b. The capsule volume (*p* < 0.01) and ratio (*p* < 0.05) in the left adrenal were significantly higher than the right adrenal, while the cortex gland ratio and zona fasciculata (zF) cortex ratio, which are responsible for steroid hormone production in the right adrenal, were higher than the left adrenal. Figure 1c shows the lipid storage in the left and right adrenal glands. The left adrenal presents slightly higher lipid storage, but it is not statistically significant.

### 2.2. Comparison of the Major Markers of Adrenal Zones and the Subcellular Structure in the Left and Right Adrenal Glands

To compare the different zones in the bilateral adrenal gland, immunohistology staining of the major adrenal zones was conducted. The expression of Star and Cyp11b1 genes were used as the markers to represent the cortex and zona fasciculata, while the expression of Th and Pnmt genes were used to locate the medulla and mature chromaffin cells, respectively. As shown in Figure 2a,b, the Star gene was expressed throughout the cortex, while the Cyp11b1 gene expression was restricted to the zona fasciculata region. In the adrenal medulla, the Th gene was expressed throughout the medulla, while the expression of the Pnmt gene was restricted in adrenalin-producing chromatin cells (Figure 2c,d). To compare the staining potency of major markers in adrenal zones, the histoscore (H-score) of different markers in the left and right adrenal immunohistochemical staining slides were obtained. As shown in Figure 2i, immunostaining of the adrenal cortex marker Star and zF marker Cyp11b1 was higher in the right adrenal, while the H-score of adrenal medulla marker Th and Pnmt was slightly higher in the left adrenal than the right, suggesting a higher level of glucocorticoids production in the right adrenal than the left one. 

### 2.3. Differentially Expressed Genes (DEGs) in the Left and Right Adrenal Glands

Based on a detailed analyses with an electron microscope, the cell structure of zF on both left and right adrenal glands was clear with a homogeneous cytoplasm, intact cell membrane, and abundant organelles. In the left adrenal gland, the mitochondria were abundant, the matrix was uniform, the cristae were long and well arranged; occasionally the mitochondria were slightly swollen, and the cristae were somewhat broken. In the right adrenal gland, a few mitochondria were found to be slightly swollen, and the matrix was not uniform. In addition, the rough endoplasmic reticulum and smooth endoplasmic reticulum in the zona fasciculata cells in the left and right adrenal glands were not similar, while lipid droplets in both glands were abundant (Figure 2e,f). To compare the energy metabolism in two sides, the mitochondria size and number were analyzed, and a significantly greater number of mitochondria was quantified in the left adrenal than the right one (*p* < 0.05) (Figure 2g,h).

#### 2.3.1. DEGs Involved in the Capability of Glucocorticoid Production

To elucidate the sided differences in the adrenal glands at the transcriptome level, RNA sequencing was performed. As a result, a total of 2581 differentially expressed mRNAs, including 1264 up-regulated genes and 1317 down-regulated genes, were identified under the threshold of fold change > 1.2 and FDR < 0.001 (DEGs were right vs left). A hierarchical dendrogram and a PCA plot of the biological replicates are shown in Appendix A. At first, DEGs between the left and right adrenal glands were analyzed by KEGG pathway enrichment and GO enrichment. By KEGG pathway enrichment, a total of 18 pathways were significantly enriched (*p* < 0.05) (Figure 3a), of which 11 pathways belonged to the “Organismal system”, and “Corticosterone synthesis and secretion” was the most significantly up-regulated pathway in these pathways, which suggests the predominance of the right adrenal gland in the production of glucocorticoids (Table 2). DEGs related to “Corticosterone synthesis and secretion” were exhibited in a heat map in Figure 3b, including twelve significantly up-regulated genes: *Agt*, *Agtr1a*, *Agtr1b*, *Creb3*, *Cyp11b2*, *Hsd3b1*, *Hsd3b6*, *Mc2r*, *Mrap*, *Nr0b1*, *Plcb1*, and *Star*, and six significantly down-regulated genes: *Adcy1*, *Adcy3*, *Adcy6*, *Cacna1i*, *Creb3l2*, and *Kcnk3*. The fold change in these genes in the left and right adrenals were summarized in Table 3. Further, the expression of key genes involved in steroid hormone production, including *Hsd3b1*, *Cyp11b2*, *Cyp11b1*, *Cyp11a1*, and *Star*, was validated by RT-qPCR, and a significantly higher expression of steroid hormone production genes was quantified in the right adrenal (Figure 4a–e). 

#### 2.3.2. DEGs Involved in Protein Synthesis and Energy Expenditure

By analyzing significantly enriched pathways on two sides, significantly down-regulated pathways “Ribosome” and “Oxidative phosphorylation” were enriched, which belong to “Genetic Information Processing” and “Metabolism”, respectively. The pathway “Ribosome” enriched five up-regulated genes and fifty-nine down-regulated genes, while the pathway “Oxidative phosphorylation” enriched three up-regulated genes and thirty-two down-regulated genes (Table 4). Significant enrichment of these two pathways suggested that anasymmetrical protein metabolism and energy expenditure in the left and right adrenal may accord with a greater mitochondria number observed in the left adrenal cells than the right side. This imbalance in energy metabolism between the left and right adrenals was most likely related to a higher organ weight in the left adrenal gland and is associated with a higher incidence of tumors, as seen in other paired organs [16,17]. Subsequently, protein–protein interaction (PPI) networks were constructed with significantly differently expressed genes, and three major PPI networks along with ten hub genes in each PPI network were identified. Then, KEGG pathway enrichment indicated that these networks were responsible for “Ribosome”, “Complement and coagulation cascades”, and “Oxidative phosphorylation”, respectively (Figure 3d–f). These data suggest that there is a difference in energy metabolism and protein synthesis in the left and right adrenal glands.

#### 2.3.3. DEGs Involved in Different Neural Innervation and Adrenal Regeneration

By GO enrichment, the top ten significantly enriched GO terms with the lowest *p*-value in three GO classifications (BP, CC, MF) were identified (Appendix A), and GO:0007399: nervous system development was the most significantly enriched GO term in the biological process category, which suggested asymmetrical neural innervation in the left and right adrenals. Further, seven downstream GO terms of GO:0007399: nervous system development were identified (Appendix A), and nineteen hub genes, which include *Atp8a2*, *Ccl5*, *Cend1*, *Cit*, *Cma1*, *Cxcl1*, *Cyp46a1*, *Emx1*, *Fgf10*, *Gjb1*, *Hoxd9*, *Lst1*, *Mt3*, *Nrn1*, *Ppargc1a*, *Rgs14*, *Rxrg*, *Sf3a2*, and *Trim67*, were identified in a chord graph (Figure 3c). By investigating the role of laterality genes in asymmetric adrenal development, a significantly higher expression of *lefty-2* in the left adrenal and a higher expression of *Shh* in the right adrenal were observed. Given the important role of Shh signaling in adrenal stem cell maintenance, it is assumed that Shh signaling may play an important role in the laterality difference in adrenal glands. Further, RT-qPCR was performed, and a significantly higher expression of *Shh* in the right adrenal was quantified (Figure 4f). In addition, dispatched 1 (*Disp1*) and its paralog *Disp2*, a multi-pass transmembrane protein required for the release of cholesterol-modified active Shh (ShhNp) from the generating cells, were also significantly differentially expressed in the left and right adrenal glands [18,19]. 

#### 2.3.4. Construction of the ceRNA Network and Hub miRNA Identification

In addition to the 2581 differentially expressed mRNAs, 56 differentially expressed LncRNAs and 132 differentially expressed miRNAs were identified by whole transcriptome sequencing under the threshold of a fold change > 2 and a significance of *p* < 0.001 and FDR < 0.05, respectively. According to the predictions of binding sites and expression relationships among miRNAs, lncRNAs, and mRNAs, a competing endogenous RNA network comprising eight differentially expressed LncRNAs, twenty-three miRNAs, and fourteen mRNAs was constructed (Figure 5a). Further, the most significant sub-ceRNA network composed of hub miRNA miR466i-5p, two LncRNA ENSMUST00000197424 and ENSMUST00000215724, and seventy-three mRNAs was identified (Figure 5b). Functional enrichment indicated that the predicted target genes of miR466i-5p including *Epm2a*, *Ngfr*, *Dcdc2a*, *Tox*, *Chrdl1*, and *Elavl3* were responsible for nervous system development. The expression of miR466i-5p along with its target genes, including *Ngfr*, *Cend1*, *Cdh6*, *Gata2,* and *Tox,* was validated by RT-qPCR (Figure 6a–f), and significantly different expressions of these genes in the left and right adrenals were observed. 

## 3. Discussion

In the present study, we identified differently expressed genes in the left and right adrenal glands and analyzed the genetic bases responsible for anatomical, physiological, and functional differences in the two sides. The first difference identified in the left and right adrenal was the capacity of glucocorticoid production. Glucocorticoids are the major steroids produced from the adrenal cortex, which regulates metabolism, development, and immune function in the body, and the dysregulation of glucocorticoid production is associated with metabolic diseases, such as diabetes and even psychiatric disorders [14,20]. By KEGG pathway enrichment for differently expressed genes between the left and right adrenal glands, the pathway “Corticosterone synthesis and secretion” was significantly enriched. The genes in this pathway were more expressed in the right adrenal gland than in the left one, suggesting more abundant glucocorticoid production in the right adrenal glands. Furthermore, we compared the volume and cortex ratio of zona fasciculata in the left and right adrenal glands, which is responsible for glucocorticoid production. Although the volume of zona fasciculata was not statistically different in the two sides, possibly due to the limited sample size, the cortex ratio of zona fasciculata was larger in the right gland than the left side. In addition, a higher expression of zF marker Cyp11b1 in the right adrenal was observed by immunostaining, and the mRNA expression levels of multiple key genes involved in steroid hormone production were significantly expressed to a greater extant in the right adrenal than the left one. Therefore, the right adrenal gland is dominant for corticosterone synthesis and secretion in adult male mice. 

It has been suggested that the asymmetry in glucocorticoid production is age-dependent, with right-sided dominance in young animals but no difference in older animals [21]. This age dependence of adrenal gland characteristics may be related to compensatory adrenal growth. In the adrenal gland, unilateral adrenalectomy can increase the compensatory growth and steroidogenesis in the remaining adrenal gland [22]. Thus, when mono-lateral adrenal function decreased with age, the differences in bilateral adrenal function may be reduced. There is a limitation in the present study. The actual hormone secretion of bilateral adrenal glands in the present study was not detected due to the low organ weight and low steroid hormone production of the adrenals in mice. 

The second difference in the left and right adrenal glands was their neural innervation and adrenal regeneration. Adrenal innervation plays a critical role in the regulation of steroid secretion and catecholamine release in the gland and exerts regulatory actions on compensatory adrenal growth, diurnal rhythm of plasma corticosterone, and regeneration [23,24,25]. Side-specific adrenal innervation from the central structure may lead to completely different molecular activity in the bilateral adrenals, further resulting in diverse functional activities [12,14]. By GO functional enrichment for differently expressed genes in the left and right adrenal glands, GO:0007399: nervous system development was significantly enriched, and seven low-layer GO terms, along with common genes, were identified. In addition, GO:0048485: sympathetic nervous system development which is crucial for catecholamine secretion from the adrenal medulla and facilitates cell communication within the adrenal cortex and adrenal medulla, was significantly enriched in two sides. Furthermore, the various genes responsible for nervous system development were validated by RT-qPCR, and a higher expression of these genes in the left adrenal than the right adrenal was observed. These results corroborated the previous findings that nervous innervations in the left and right adrenal glands are different. Side-specific nervous innervation might exist between the left and right adrenal glands that control different adrenal functions of the left and right adrenal glands.

We investigated the role of laterality genes in the asymmetric development of bilateral adrenal glands and identified *Shh* as the most critical gene for asymmetric adrenal growth. Shh signaling is a major regulator of embryo development. It is involved in the regulation of CNS Polarity [26] and regulates the development and maintenance of the adrenal capsule and cortex [27]. In the present study, Shh is significantly more expressed in the right adrenal gland than in the left, which is consistent with previous findings that the right adrenal gland is dominant for steroid production. A previous study reported that the mice carrying Shh^fl/fl;SF-1/Cre+^ made only the left adrenal gland detectable, while the right adrenal gland was absent [28]. Another study demonstrated that the conditional deletion of *Shh* in steroidogenic cells in mouse adrenals affected the left and right adrenals differently [29]. These observations provided additional evidence to support the hypothesis that Shh signaling significantly contributes to the asymmetries of the left and right adrenal glands, both in terms of morphology and function. 

In vertebrates, the anatomical and physiological differences in the paired organs, including tissue volume, structure, location, arterial supply, and venous and lymphatic drainage, are precisely regulated during embryonic development [30]. Secreted growth factors involving *nodal*, *lefty-1,* and *lefty-2* play crucial roles in establishing left–right asymmetries [31,32], and *Pitx2* is responsible for asymmetries in internal organs, such as the heart and gastrointestinal tract [33]. In the present study, the laterality gene lefty-1 was significantly more expressed in the left adrenal. There is a potential link between the differential expression of laterality genes and the occurrence of cancer [34]. More importantly, patient survival rates were different according to the primary status of the tumor [30,35]. In the present study, more massive left adrenals were detected, and the genes for protein synthesis and energy metabolism were highly expressed in the left adrenals. Asymmetries in morphology and function in the left and right adrenals may lead to a higher incidence of unilateral adrenal adenomas on the left side compared to the right. In cancers of other paired organs, including the breast, kidney, and ovary, similar lateralizing asymmetries were also observed, leading scientists to consider laterality as an independent prognostic factor in these cancers [36,37,38]. 

In conclusion, we have investigated the anatomical and physiological differences in the left and right adrenal glands and analyzed the differently expressed genes on either side. We have identified the key genes responsible for asymmetric neural innervation, energy expenditure, and glucocorticoid production, with particular emphasis on the *Shh* gene and miRNA-466i-5p.

## 4. Materials and Methods

### 4.1. Animal Experiment

Male C57BL/6J mice (*n* = 20) at the age of 11 weeks, weighing 25–28 g, were purchased from the Model Animal Research Center of Nanjing University (Nanjing, China). The animals were maintained at 22 ± 2 °C with a 12 h light/dark cycle and given free access to standard chow and water for a week at the Animal Experimental Center of China Pharmaceutical University (CPU). The animals were sacrificed with cervical dislocation after being anesthetized with 4% isoflurane (RWD Life Science, Shenzhen, China). The bilateral adrenals were then harvested, weighed, and immersed in RNA (Servicebio, Wuhan, China). Animal experiments in this study complied with the China Pharmaceutical University Guide for Laboratory Animals and the ARRIVE guidelines. The Institutional Animal Care and Concern Committee at China Pharmaceutical University approved all experimental protocols.

### 4.2. Adrenal Gland Histology and Oil Red O Staining

For histological studies, the adrenal glands (*n* = 6) were fixed in 4% paraformaldehyde (PFA) for 24 h before paraffin embedding, cut into 5 μm thick sections, and stained with hematoxylin and eosin (H&E). Mosaic images were taken using the digital whole slide pathology scanner Leica Aperio VESA8 (Leica Biosystems, Wetzlar, Germany). The three largest equatorial sections of each adrenal gland were obtained, and the areas of the entire gland, zona glomerulosa, zona fasciculate, and medulla were quantified using Image J Fiji software (version 1.53q, Wayne Resband and contributors, National Institutes of Health, Rockville, MD, USA) [39]. The area of the adrenal cortex was calculated by subtracting the capsule and medullary from the total adrenal area. The thickness of the zona fasciculata was calculated by subtracting zona glomerulosa from the cortex because the male mouse has no recognizable zona reticularis [40]. By analyzing the volume of different compartments, the ratio of capsule, cortex, and medulla throughout the gland, as well as the ratio of zona glomerulosa and zona fasciculata throughout the cortex, were calculated.

Cholesterol is the precursor to produce steroid hormones, and excess cholesterol can be stored as cholesterol esters in lipid droplets [41]. To assess the intracellular storage of cholesterol, bilateral adrenals (*n* = 4) were harvested and immediately stored in liquid nitrogen before oil red O staining of the lipid droplets. A series of 5 μm cryo-sections of the adrenal glands were fixed in 4% PFA for 1 min and then rinsed in 60% isopropyl alcohol for 3 min. Then, the sections were stained by an oil red O staining solution (Sigma, Darmstadt, Germany) for 15 min and counterstained with hematoxylin. The microscopic images were taken using a Leica HS6 digital whole slide pathology scanner (Leica Microsystems, Wetzlar, Germany), and the area of lipid droplets (pixels) in the cortex was quantified using Image J Fiji software.

### 4.3. Transmission Electron Microscopy (TEM) and Immunohistochemistry

To detect the subcellular structure in the left and right adrenals, transmission electron microscopy was performed. The adrenals (*n* = 3) were taken within 1 to 3 min, and their surrounding fat tissues were removed. The size of the tissue block was restricted to no more than 1 mm3, and fixation was conducted by immersing the sample in 2.5% glutaraldehyde for 2 h at room temperature and 12 h at 4 °C. The tissues were washed with 0.1 M PB (pH 7.4), dehydrated in ethanol, embedded in LR white resin, and then stained with uranyl acetate and lead citrate. The ultrathin sections (80 nm) were cut on an ultra-microtome (Leica UC7; Leica Microsystems, Wetzlar, Germany), and images were examined by a transmission electron microscope (HT7800; Hitachi High-Technologies, Tokyo, Japan). The low-magnified TEM images were used to analyze the mitochondria number, and high-magnified TEM images were used to measure the mitochondria size in adrenal cells. The image analysis was performed using Image Pro-Plus 6.0 software (Media Cybernetics, Inc., Rockville, MD, USA). For immunohistochemistry, paraffin-embedded adrenal slides (*n* = 5) were deparaffinized in xylene, hydrated in graded ethanol solutions, and incubated with an AR buffer before heat-induced antigen retrieval was performed. To block endogenous peroxidases and antigens, 3% H_2_O_2_ and 3% bovine serum albumin were used. Briefly, multiple rounds of staining containing incubation with a primary antibody, staining with a horseradish peroxidase-conjugated secondary antibody, and Opal fluorescent substrate deposition were performed. At last, the nuclei were counterstained with DAPI, and images of the slides were obtained by a Leica Aperio VESA8 scanner. Digital images of immunohistochemistry slides were analyzed using a whole slide image analysis platform, HALO software, version 3.4.2986 (Indica Labs, Albuquerque, NM, USA). Immunoreactive cells were identified, and the histoscore (H-score, possible range from 0 to 300) was calculated by a semi-quantitative assessment of both the intensity of staining and the percentage of positive cells. The antibodies were purchased from Proteintech Group (Rosemont, IL, USA ), Santa Cruz Biotechnology (Dallas, TEX, USA), Abcam (Waltham, MA, USA), or GeneTex (Irvine, CA, USA): rabbit anti-STAR antibody (1:200, 12225-1-AP, Proteintech), mouse anti-CYP11B1 antibody (1:50, sc-377401, Santa Cruz), rabbit anti-TH antibody (1:500, ab137869, Abcam), and rabbit anti-PNMT antibody (1:400, GTX114098, GeneTex). For the positive and negative control in IHC, mouse cerebral cortex tissue (anti-EP1532Y antibody and anti-PNMT antibody), mouse kidney (anti-Cyp11b1 antibody), and mouse testis tissue (anti-Star antibody) were used. PBS was employed as the blank control in place of the primary antibody.

### 4.4. RNA Extraction and RNA Sequencing Analysis

Total RNA was extracted from bilateral adrenal glands using TRIZOL Reagent (Invitrogen, Carlsbad, CA, USA), according to the manufacturer’s instructions. The quality of RNA samples was assessed by a NanoPhotometer^®^ spectrophotometer (IMPLEN, Munich, Germany), and the RNA concentration was measured using a Qubit^®^ 2.0 Fluorometer (Life Technologies, Carlsbad CA, USA). The RNA Integrity Number (RIN) was then determined using an Agilent Bioanalyzer 2100 system (Agilent Technologies, Palo Alto, CA USA). After removing the ribosomal RNA, a TruseqTM RNA sample prep Kit (Illumina, San Diego, CA, USA) was used to construct sequencing libraries for lncRNA, circRNA, and mRNA sequencing, and a Truseq TM Small RNA sample Prep Kit (Illumina) was used for miRNA sequencing. Then, clustering generation was performed using a TruSeq SR Cluster Kit v3-cBot-HS (Illumina), and the subsequent sequencings (pair-end 150 bp for mRNAs, and single-end 50 bp for miRNAs) were conducted on a respective Illumina NovaSeq 6000 platform, following the manufacturer’s recommendations. The clean data were used for downstream analyses, and the expression level for each transcript was calculated using the transcripts per million mapped reads (tpm) method [42]. The differential expression analysis was performed using the R statistical package edge R [43]. The figures of RNA sequencing results were generated using an online data analysis and visualization platform (http://www.bioinformatics.com.cn/, accessed on 28 April 2022). 

### 4.5. Functional Analysis and PPI Network Establishment

Gene Ontology (GO) analysis and Kyoto Encyclopedia of Genes and Genomes (KEGG) pathway analysis were performed to investigate the biological functions of differentially expressed genes in the left and right adrenal gland using Goatools software (https://github.com/tanghaibao/GOatools, accessed on 28 April 2022) and the KOBAS 3.0 online tool (http://kobas.cbi.pku.edu.cn/kobas3, accessed on 28 April 2022) [44,45]. The GO annotation is composed of three functional terms: biological processes (BPs), cellular components (CCs), and molecular functions (MFs). After an analysis of the significance level (*p*-value) and false discovery rate (FDR) of GO terms, enriched gene sets in significant GO terms (*p*-value < 0.05) were investigated. A KEGG pathway analysis was performed to explore pathway clusters responsible for molecular interaction and reaction networks of the differentially expressed genes, and a *p*-value < 0.05 was considered as a significant pathway in the present study. The protein–protein interaction (PPI) network was constructed by the STRING (Search Tool for the Retrieval of Interacting Genes) database and visualized by Cytoscape software. The “Molecular Complex Detection” (MCODE) clustering plugin was used to recognize densely connected regions in a large PPI network, and hub genes in each PPI network were selected using the cytoHubba plugin [46].

### 4.6. Construction of the ceRNA Regulatory Network

RNA molecules, such as lncRNAs and mRNAs, can communicate with each other by sharing miRNA response elements (MREs), according to the ceRNA theory [47] (Salmena, et al., 2011). To investigate the role and interaction among differentially expressed mRNAs, lncRNAs, and miRNAs in asymmetric adrenal development, a competitive endogenous RNA (ceRNA) network was constructed based on the targeting relationship between mRNA/lncRNA and microRNA. The miRanda (http://www.microrna.org/microrna, accessed on 15 May 2022) database was used to predict the putative targets for lncRNAs and mRNAs with default parameters (score > 165, E value < −25), and Cytoscape v3.9.0 was used to visualize the lncRNA–miRNA–mRNA interaction network and major sub-network.

### 4.7. RT-qPCR

The expression of multiple genes involved in steroid hormone production, including the mRNA levels of *Hsd3b1*, *Cyp11b2*, *Cyp11b1*, *Cyp11a1*, and *Star*, and the genes involved in nervous system development, including *Ngfr*, *Cend1*, *Cdh6*, *Gata2*, and *Tox*, were analyzed by RT-qPCR. The specificity and efficiency of primers were checked with melt curve analysis, and the 2^−ΔΔCt^ method was used to process the data in the experiment. *Gapdh* was chosen as an endogenous control to normalize the expression levels of mRNA, while U6 was chosen as an endogenous control to normalize the expression levels of miR-466i-5p. The primer sequences are listed in Table 5.

### 4.8. Statistical Analysis

The Kolmogorov–Smirnov test was used to evaluate the data distribution. Adrenal weight and body weight ratio data were analyzed with an independent Mann–Whitney *U*-test. Statistical analyses of histological parameters, immunohistochemical staining, mitochondria parameters, and RT-qPCR validations were performed using unpaired Student’s *t*-test (after testing for normality with the F test), and *p* < 0.05 was considered statistically significant. Experimental data were expressed as the mean ± SEM. Statistical graphs were generated using the GraphPad Prism 5 software.

## Figures and Tables

**Figure 1 ijms-24-17456-f001:**
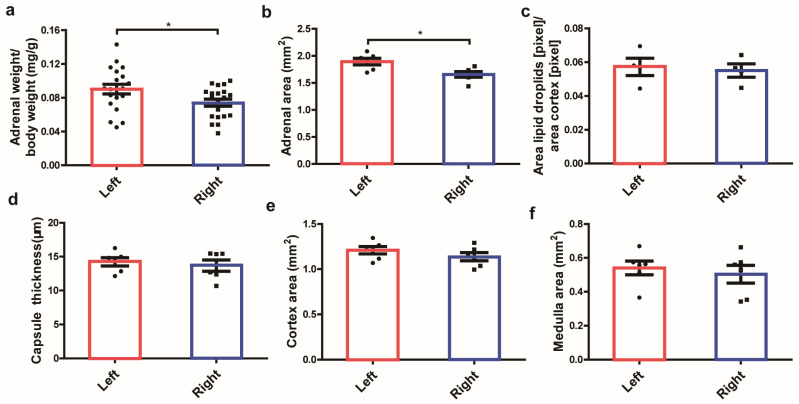
Weight, volume, and lipid storage capacity between the left and right adrenal glands in mice. Organ weight (**a**), volume (**b**), lipid storage capacity (**c**), capsule thickness (**d**), the area of two main components cortex (**e**), and the medulla (**f**) in left and right adrenal glands in H&E staining. * *p* < 0.05, Mann–Whitney *U*-test.

**Figure 2 ijms-24-17456-f002:**
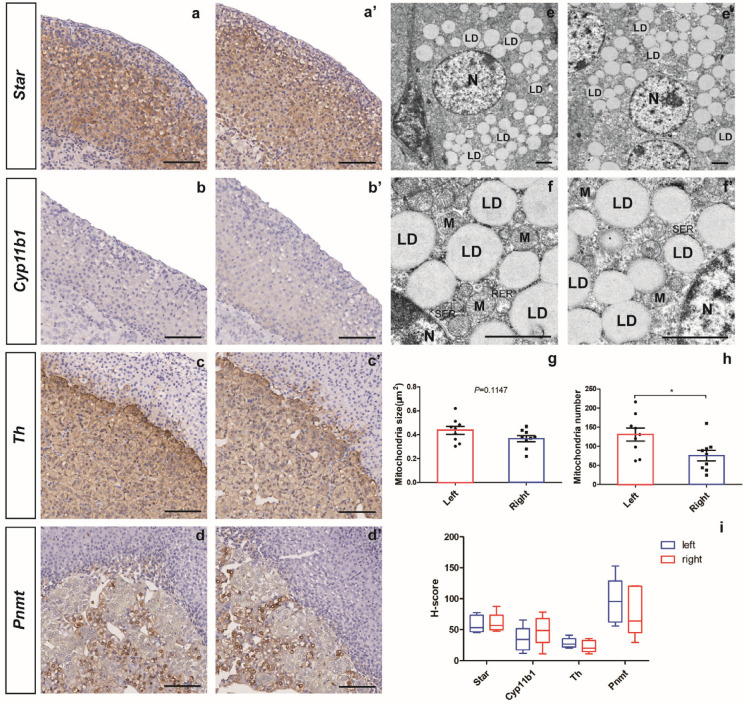
Comparison of major markers in the adrenal zones and subcellular structure under TEM between the left and right adrenal glands in mice. Immunohistochemistry of major adrenal markers, including Star (**a**,**a’**), Cyp11b1 (**b**,**b**’), Th (**c**,**c’**), and Pnmt (**d**,**d’**), and H-score analysis of the adrenal markers (**i**). Low (**e**,**e’**) to high (**f**,**f’**) magnified images of the adrenal cortex under transmission electron microscopy (TEM) and the analysis of mitochondria size (**g**) and mitochondria number (**h**). N = nucleus, M = mitochondria, LD = lipid droplets, RER = rough endoplasmic reticulum, SER = smooth endoplasmic reticulum. Scale bars are 100 μm for immunohistochemistry and 2 μm for TEM. * *p* < 0.05, *t*-test.

**Figure 3 ijms-24-17456-f003:**
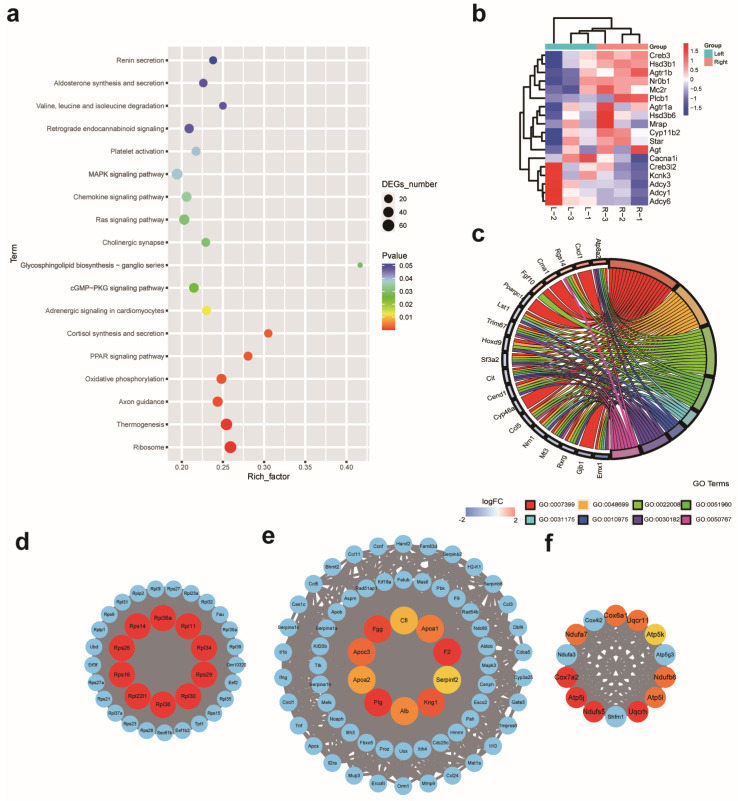
Transcriptional sequencing analyses between the left and right adrenal glands in mice. Significantly enriched KEGG pathways (**a**), heatmap of DEGs in the KEGG pathway “Corticosterone synthesis and secretion” (**b**), and chord plot containing significantly enriched GO terms related to nervous system development (**c**). Major protein–protein interaction network and top 10 hub genes in each network (**d**–**f**). The node color changed gradually from yellow to red in ascending order, according to the maximal clique centrality of the node.

**Figure 4 ijms-24-17456-f004:**
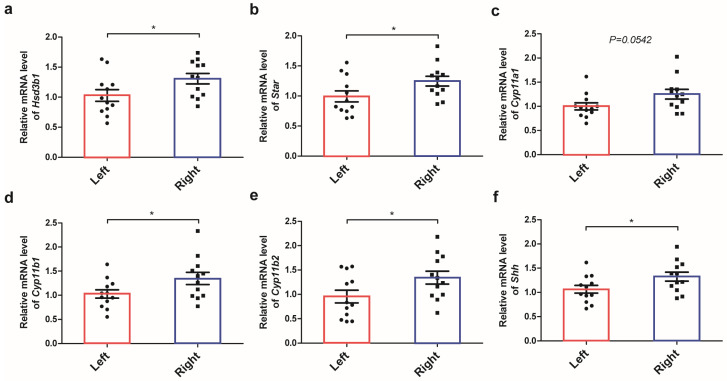
Relative expression of multiple genes involved in steroidogenesis. Real-time qPCR quantification of the mRNA expression of multiple genes affecting steroid hormone production in adrenal glands, including *Hsd3b1* (**a**), *Star* (**b**), *Cyp11a1* (**c**), *Cyp11b1* (**d**), and *Cyp11b2* (**e**). The mRNA expression in the adrenal cortex progenitor cell marker *Shh* (**f**). * *p* < 0.05, *t*-test.

**Figure 5 ijms-24-17456-f005:**
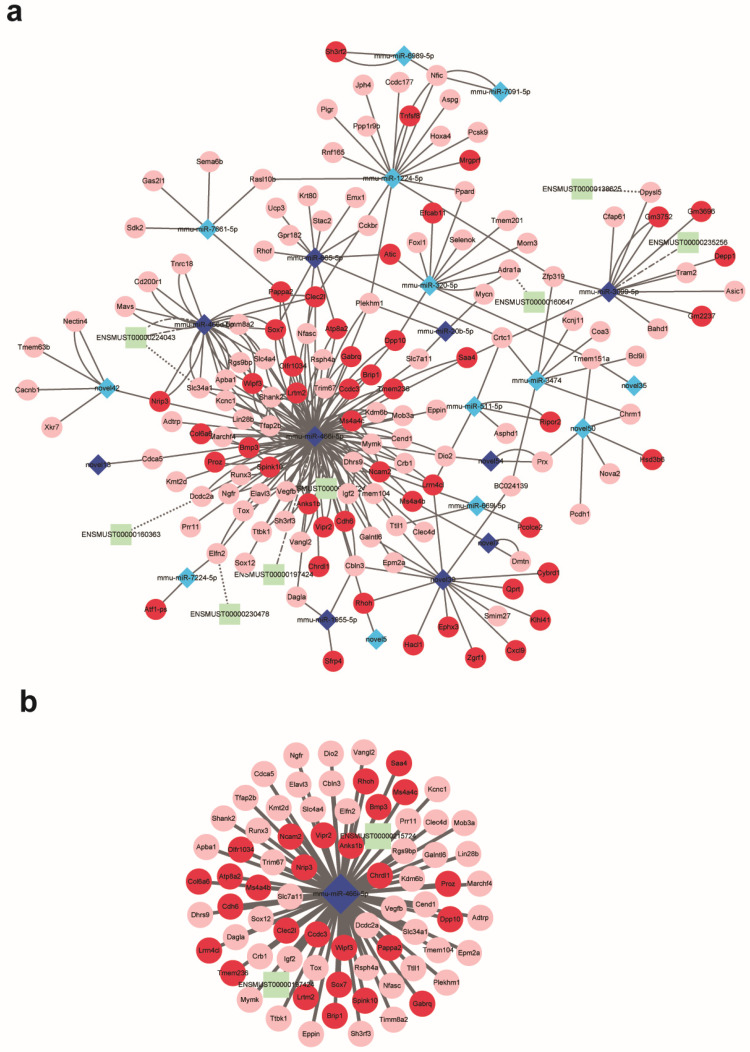
Construction of the ceRNA regulatory network. The whole competing endogenous RNA network (**a**) and the sub-network consists of miRNA-466i-5p, two LncRNAs, and seventy-three mRNAs (**b**). Up- and down-regulated LncRNAs, miRNAs, and mRNAs were exhibited in dark and light green and blue and red, respectively. Solid lines indicate the relation between miRNAs and mRNAs, and dotted lines indicate the relation between LncRNAs and mRNAs.

**Figure 6 ijms-24-17456-f006:**
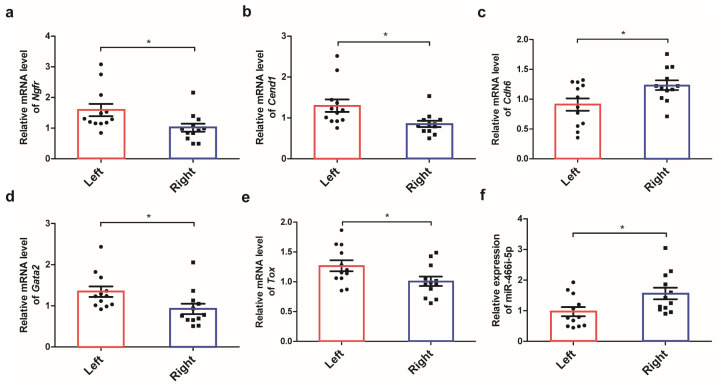
The relative expression of various genes involved in nervous system development and hub micro-RNA. Real-time qPCR quantification of the mRNA expression of multiple genes involved in the nervous system development of adrenal glands, including *Ngfr* (**a**), *Cend1* (**b**), *Cdh6* (**c**), *Gata*2 (**d**), and *Tox* (**e**), and the expression of hub microRNA miR-466i-5p (**f**). * *p* < 0.05, *t*-test.

**Table 1 ijms-24-17456-t001:** (**a**) Morphometric analysis of the areas of different compartments in the left and right adrenals. (**b**) Morphometric analysis of the ratio of different compartments in the left and right adrenals.

**(a)**			
**Areas**	**Left Adrenal**	**Right Adrenal**	***p*-Value**
Entire gland	1.867 ± 0.155 ^a^*	1.691 ± 0.147	0.049
Capsule	0.081 ± 0.019 ^b^**	0.052 ± 0.006	0.005
Cortex	1.210 ± 0.102	1.138 ± 0.112	0.272
zG	0.267 ± 0.046	0.243 ± 0.037	0.343
zF	0.943 ± 0.060	0.895 ± 0.091	0.307
Medulla	0.540 ± 0.10	0.503 ± 0.129	0.589
**(b)**			
**Ratio**	**Left Adrenal**	**Right Adrenal**	***p*-Value**
Capsule/gland	0.044 ± 0.012 ^a^*	0.031 ± 0.003	0.021
Cortex/gland	0.661 ± 0.041	0.674 ± 0.058	0.681
zG/Cortex	0.219 ± 0.022	0.213 ± 0.022	0.645
zF/Cortex	0.781 ± 0.022	0.787 ± 0.022	0.645
Medulla/gland	0.294 ± 0.045	0.295 ± 0.061	0.976

Data were means ± SEM. All measurements are in mm^2^ (square millimeters). The number of mice is 6. ^a^ A significantly (* *p* < 0.05, unpaired *t*-test) higher value compared to the contralateral adrenal gland. ^b^ A significantly (** 0.001 < *p* < 0.01, unpaired *t*-test) higher value compared to the contralateral adrenal gland.

**Table 2 ijms-24-17456-t002:** Top 10 pathways belonging to “Organismal Systems” in the significant enrichment of 18 classical pathways.

Pathway	Level 1	Level 2	ID	Up/Down Number	DEG Number	Total Number	*p*-Value
Thermogenesis	Organismal Systems	Environmental adaptation	ko04714	11/48	59	232	1.92 × 10^−5^
Axon guidance	Organismal Systems	Development and regeneration	ko04360	13/25	38	156	0.00129021
PPAR signaling pathway	Organismal Systems	Endocrine system	ko03320	8/15	23	82	0.001617661
Corticosterone synthesis and secretion	Organismal Systems	Endocrine system	ko04927	12/6	18	59	0.00180696
Adrenergic signaling in cardiomyocyte	Organismal Systems	Circulatory system	ko04261	13/16	29	126	0.010343082
Cholinergic synapse	Organismal Systems	Nervous system	ko04725	6/16	22	96	0.02441011
Chemokine signaling pathway	Organismal Systems	Immune system	ko04062	18/17	35	170	0.028186015
Platelet activation	Organismal Systems	Immune system	ko04611	12/13	25	115	0.031805019
Retrograde endocannabinoid signaling	Organismal Systems	Nervous system	ko04723	5/23	28	134	0.038658635
Aldosterone synthesis and secretion	Organismal Systems	Endocrine system	ko04925	10/9	19	84	0.039170725

**Table 3 ijms-24-17456-t003:** Differentially expressed genes in the KEGG pathway “Corticosterone synthesis and secretion” between the left and right adrenal glands.

Gene Symbol	KEGG	Description	Fold Change (L/R)
*Plcb1*	K05858	1-phosphatidylinositol 4,5-bisphosphate phosphodiesterase beta-1 isoform X2	1.27
*Agtr1b*	K04166	type-1B angiotensin II receptor	1.31
*Hsd3b1*	K00070	3 beta-hydroxysteroid dehydrogenase/Delta 5--4-isomerase type 1	1.50
*Hsd3b6*	K00070	3 beta-hydroxysteroid dehydrogenase/Delta 5--4-isomerase type 6	1.64
*Creb3*	K09048	cyclic AMP-responsive element-binding protein 3	1.33
*Kcnk3*	K04914	potassium channel subfamily K member 3	0.77
*Creb3l2*	K09048	cyclic AMP-responsive element-binding protein 3-like protein 2	0.76
*Agt*	K09821	angiotensinogen isoform X1	1.94
*Star*	K16931	steroidogenic acute regulatory protein, mitochondrial progenitor	1.20
*Adcy1*	K08041	adenylate cyclase type 1	0.75
*Adcy3*	K08043	adenylate cyclase type 3 isoform 2	0.82
*Agtr1a*	K04166	type-1A angiotensin II receptor isoform X1	1.23
*Adcy6*	K08046	adenylate cyclase type 6 isoform 1	0.77
*Cacna1i*	K04856	voltage-dependent T-type calcium channel subunit alpha-1I isoform X2	0.61
*Cyp11b2*	K00497	cytochrome P450 11B2, mitochondrial	1.36
*Mrap*	K22398	melanocortin-2 receptor accessory protein	1.23
*Mc2r*	K04200	adrenocorticotropic hormone receptor	1.21
*Nr0b1*	K08562	nuclear receptor subfamily 0 group B member 1	1.64

**Table 4 ijms-24-17456-t004:** Significantly enriched pathways not belonging to “Organismal Systems”.

Pathway	Level 1	Level 2	ID	Up/Down NO.	DEG NO.	Total NO.	*p*-Value
Ribosome	Genetic Information Processing	Translation	ko03010	5/59	64	247	4.54 × 10^−6^
Oxidative phosphorylation	Metabolism	Energy metabolism	ko00190	3/32	35	141	0.001383233
cGMP-PKG signaling pathway	Environmental Information Processing	Signal transduction	ko04022	14/18	32	149	0.019947882
Glycosphingolipid biosynthesis-ganglio series	Metabolism	Glycan biosynthesis and metabolism	ko00604	2/3	5	12	0.023467218
Ras signaling pathway	Environmental Information Processing	Signal transduction	ko04014	17/23	40	197	0.024806251
MAPK signaling pathway	Environmental Information Processing	Signal transduction	ko04010	20/28	48	247	0.031550245
Valine, leucine, and isoleucine degradation	Metabolism	Amino acid metabolism	ko00280	8/5	13	52	0.039016313

**Table 5 ijms-24-17456-t005:** Primers used for RT-qPCR.

Gene Symbol	NCBI Accession No.	Product Size	Primer Sequence (5′ to 3′)
*Hsd3b1*	NM_001304800.1	85	TGCTGCACAGCCCTCCTA
			TCCATCCAGCCATGGTCAAC
*Cyp11b2*	NM_009991.4	251	TGGCATTGTGGCGGAACTAA
			AAGGGGATTGCTGTCGTGTC
*Cyp11b1*	NM_001033229.3	162	CGCTGCAAATCCTCAGAAGG
			ACATTGAGGACTGTCCCAGCA
*Cyp11a1*	NM_001346787.1	189	CCCGGAGAGCTTGTGCAAAT
			CCCATGCTGAGCCAGATGTC
*Star*	NM_011485.5	184	TCGCTACGTTCAAGCTGTGT
			GCTTCCAGTTGAGAACCAAGC
*Ngfr*	NM_033217.3	102	CCCTGCCTGGACAGTGTTAC
			ACAGGGAGCGGACATACTCT
*Cend1*	NM_001360485.1	159	CAGGACGGGGAGACCCAT
			CCTTGGTATCTGGCTTGGGG
*Cdh6*	NM_007666.4	106	CAGCAAGAAGCACAGAGCAG
			CTTTCAGAGGGTACCTCGGTT
*Tox*	NM_001377078.1	142	TGGTCAGCTGCACACTAGAA
			ACTCCTTTTCTTTTCTCCTGCC
*Shh*	NM_009170.3	168	ACGTAGCCGAGAAGACCCTA
			ACTTGTCTTTGCACCTCTGAGT
*Gata2*	NM_001355253.1	104	TGTCAGACGACAACCACCAC
			AGTGGCCTGTTAACATTGTGC
*Gapdh*	NM_001411840.1	166	AGGTCGGTGTGAACGGATTT
			ACTGTGCCGTTGAATTTGCC
U6	NR_138085.1	79	CTCGCTTCGGCAGCACA
			AACGCTTCACGAATTTGCGT
miR-466i-5p	NR_035412.1	65	AATCGGCGTGTGTGTGTGTGT
			ATCCAGTGCAGGGTCCGAGG

## Data Availability

Data are presented in the text, figures, and tables of this article and will be made available upon request. RNA sequencing data related with this study can be accessed in the NCBI Sequence Read Archive (SRA) database with Bioproject ID PRJNA976187 (https://www.ncbi.nlm.nih.gov/sra/PRJNA976187, accessed on 25 May 2023).

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
