# Peer review of "Asymmetries of Left and Right Adrenal Glands in Neural Innervation and Glucocorticoids Production"

_ijms, 2023, doi:10.3390/ijms242417456_

Round 1

Reviewer 1 Report

Comments and Suggestions for Authors

The manuscript presents data about the lateralization of mice adrenal glands in terms of morphology, histology, gene expression…

The study is well presented and clear. The sections are relevant.

Interestingly, the pathways generated by the study of gene expression are convergent with other data suggesting robust findings.

A few points may merit clarification / explanation / hypothesis if the authors dare to speculate.

Although likely, an enhanced gene expression of corticosterone synthesis or innervation pathways are not definite proof of lateralization of actual secretion. This merits to be mentioned.

Shh expression is lateralized. Do the authors have possible explanation such as influence of innervation, blood flow, autocrine/paracrine role of the growth factors mentioned, etc?

Are there any arguments to suggest that the origin of neural pathways towards the adrenals may influence differently one or the other adrenal? If so, the response to lateralized stimulus for instance may elicit different patterns of glucocorticoid secretion. This may have consequences in pathologies such as depression as mentioned by the authors.

Comments on the Quality of English Language

English is correct; very minor editing may be performed

Author Response

Thank you very much for your valuable comments and suggestions on our manuscript. According to the comments and suggestions, we have revised the manuscript and highlighted in blue. Herein, please find our responses point by point as below.

  1. Minor editing of English language required

Response:

Yes, we have made the modification in English edition.

  1. Although likely, an enhanced gene expression of corticosterone synthesis or innervation pathways are not definite proof of lateralization of actual secretion. This merits to be mentioned.

Response:

Thank you very much!

We fully agree with you. However, it is difficult to detect the actual level of corticosterone in mouse adrenal glands. First, left, and right adrenal glands in mouse are small. Second, the steroid hormone production of adrenal glands in mouse under physiological conditions (without stress) is low although the production under stress is increased. Therefore, in the present study, we have comparatively analyzed the volume of zF, immunostaining of zF marker and the expression of multiple genes responsible for corticosterone secretion.

The difficulties described above could be the limitations in the present study. We have included the limitations in the discussion section.

  1. Shh expression is lateralized. Do the authors have possible explanation such as influence of innervation, blood flow, autocrine/paracrine role of the growth factors mentioned etc.?

Response:

The laterality gene lefty-1 is one of the growth factors. In the present study, we found that this gene expression level was higher in left adrenal grand compare to the right. Evidence has indicated that differentially expression of laterality gene may be related with cancer incidence but no evidence regarding the role of theses growth factors in the innervation, blood flow, autocrine/paracrine function of adrenal gland was found. We have added the relevant references in the discussion section.

  1. Are there any arguments to suggest that the origin of neural pathways towards the adrenals may influence differently one or the other adrenal? If so, the response to lateralized stimulus for instance may elicit different patterns of glucocorticoid secretion. This may have consequences in pathologies such as depression as mentioned by the authors.

Response:

Adrenal function was controlled by hypothalamus via the HPA axis. Holzwarth and Dallman et al. have demonstrated that adrenal growth can be blocked by lesion on hypothalamus ipsilateral to the adrenal removed (Ref. 11). We agree with your point that lateralized stimulus in this system may result in aberrant glucocorticoid secretion and contribute to the development of mental disorders in the central nervous system. The point has been included in the discussion section.

Reviewer 2 Report

Comments and Suggestions for Authors

The manuscript titled 'Asymmetries of Left and Right Adrenal Glands in the Neural Innervation and Glucocorticoids Production', contributes valuable insights into the asymmetry of adrenal glands, but revisions are needed to enhance clarity and conciseness. Here are some comments to be considered for a more polished and impactful manuscript.

1.           The abstract provides a clear overview of the study, but the structure of the manuscript can be improved for better readability. Consider reorganizing sections to follow a more logical flow.

2.           The introduction effectively outlines the importance of left-right asymmetry in vertebrates and introduces the adrenal glands. However, it would be beneficial to include more context regarding existing research gaps in understanding the asymmetry of paired endocrine glands.

3.           The methods section is comprehensive,

Clarify the rationale for choosing C57BL/6J mice as the animal model. The description of the animals used is adequate, but it would be beneficial to provide more information about the rationale behind choosing male C57BL/6J mice, the specific characteristics that make them suitable for the study, and any potential limitations associated with this choice.

For immunohistochemistry, it would be beneficial to mention the positive and negative controls used for each antibody to validate the specificity of the staining.

Clarify the specific analyses conducted using GraphPad Prism 5, and ensure that the version used is appropriate for the analyses performed.

Clarify why specific statistical tests were chosen for each analysis. Also, mention whether corrections for multiple comparisons were applied.

4.           The results section is detailed, and the use of figures aids in understanding. However, the text could be more concise. Consider summarizing key findings in a more condensed manner. Clarify whether the observed differences in adrenal gland characteristics are age-dependent, as suggested in the discussion.

5.           The discussion effectively interprets the results and relates them to existing literature. However, it could be more concise. Focus on highlighting the key implications of the findings and their relevance to the broader field. Provide a clearer connection between the observed anatomical and physiological differences and their potential implications for clinical outcomes, especially regarding cancer incidence.

6.           The conclusion could be strengthened by summarizing the main contributions of the study more explicitly.

7.           The language is generally clear, but some sentences are complex and could be simplified for better comprehension. Avoid redundancy in the text, especially in the description of methods and results.

Comments on the Quality of English Language

The language is generally clear, but some sentences are complex and could be simplified for better comprehension. Avoid redundancy in the text, especially in the description of methods and results.

Author Response

The manuscript titled 'Asymmetries of Left and Right Adrenal Glands in the Neural Innervation and Glucocorticoids Production', contributes valuable insights into the asymmetry of adrenal glands, but revisions are needed to enhance clarity and conciseness. Here are some comments to be considered for a more polished and impactful manuscript.

  1. The abstract provides a clear overview of the study, but the structure of the manuscript can be improved for better readability. Consider reorganizing sections to follow a more logical flow.

Response:

Thank you very much for your comments and suggestions!

According to the instruction of journal, there is no structure of abstract. However, we have revised the abstract based on introduction, aim, methods, results and conclusion to improve the readability. The revision is highlighted in blue.

  1. The introduction effectively outlines the importance of left-right asymmetry in vertebrates and introduces the adrenal glands. However, it would be beneficial to include more context regarding existing research gaps in understanding the asymmetry of paired endocrine glands.

Response:

Yes, we fully agree with you. We have added the related information regarding existing research gaps in understanding the asymmetry of paired endocrine glands in the introduction.

  1. The methods section is comprehensive, Clarify the rationale for choosing C57BL/6J mice as the animal model. The description of the animals used is adequate, but it would be beneficial to provide more information about the rationale behind choosing male C57BL/6J mice, the specific characteristics that make them suitable for the study, and any potential limitations associated with this choice.

Response:

C57BL/6J mouse is widely used in physiological studies. Importantly, the molecular genome research of this mouse strain has been well completed. This is benefit for our study at molecular levels. However, the adrenal glands in adult mouse are small (only 2-3 mg), which may cause the difficulties in experiments.

For immunohistochemistry, it would be beneficial to mention the positive and negative controls used for each antibody to validate the specificity of the staining.

Response:

Yes, the samples used for positive controls included mouse cerebral cortex tissue (for anti-EP1532Y antibody and anti-PNMT antibody), mouse kidney (for anti-Cyp11b1 antibody) and mouse testis tissue (for anti-Star antibody). Substitution of primary antibodies with PBS was served as a blank control.

Clarify the specific analyses conducted using GraphPad Prism 5 and ensure that the version used is appropriate for the analyses performed.

Response:

The distribution of the data was assessed by Kolmogorov-Smirnov test, and the normality of the data was tested by F test using GraphPad Prism 5 software. The version used was appropriate for the analyses performed.

Clarify why specific statistical tests were chosen for each analysis. Also, mention whether corrections for multiple comparisons were applied.

Response:

After the analysis of data distribution, unpaired t-test was performed if the data meet normal distribution, otherwise Mann-Whitney U-test was used. No correction for multiple comparisons was applied. We have added the information in Statistical analysis.

  1. The results section is detailed, and the use of figures aids in understanding. However, the text could be more concise. Consider summarizing key findings in a more condensed manner. Clarify whether the observed differences in adrenal gland characteristics are age-dependent, as suggested in the discussion.

Response:

Thank you very much and we have made clearer in the text. In the present study, we did not study the elderly animals. However, Holzwarth et al. have demonstrated that asymmetric adrenal function was age-dependent (Ref. 21). The age-dependence of asymmetric adrenal function may be related to compensatory adrenal growth. This point has been added into the discussion section.

  1. The discussion effectively interprets the results and relates them to existing literature. However, it could be more concise. Focus on highlighting the key implications of the findings and their relevance to the broader field. Provide a clearer connection between the observed anatomical and physiological differences and their potential implications for clinical outcomes, especially regarding cancer incidence.

Response:

We accepted your comments. The discussion has been modified and improved.

  1. The conclusion could be strengthened by summarizing the main contributions of the study more explicitly.

Response:

Yes, we have strengthened the conclusion.

  1. The language is generally clear, but some sentences are complex and could be simplified for better comprehension. Avoid redundancy in the text, especially in the description of methods and results.

Response:

We have revised the whole manuscript in English.